# Does the Presence or a High Titer of Yellow Fever Virus Antibodies Interfere with Pregnancy Outcomes in Women with Zika Virus Infection?

**DOI:** 10.3390/v15112244

**Published:** 2023-11-11

**Authors:** Isa Cristina Ribeiro Piauilino, Raillon Keven dos Santos Souza, Maurício Teixeira Lima, Yanka Karolinna Batista Rodrigues, Luís Felipe Alho da Silva, Ayrton Sena Gouveia, Alexandre Vilhena da Silva Neto, Bárbara Aparecida Chaves, Maria das Graças Costa Alecrim, Camila Helena Aguiar Bôtto de Menezes, Márcia da Costa Castilho, Djane Clarys Baia-da-Silva, Flor Ernestina Martinez Espinosa

**Affiliations:** 1Programa de Pós-Graduação em Medicina Tropical (PPGMT), Universidade do Estado do Amazonas (UEA), Manaus 6904-000, Brazil; 2Fundação de Medicina Tropical Doutor Heitor Vieira Dourado, Manaus 6904-000, Brazil; 3Fundação Ezequiel Dias, Belo Horizonte 30510-010, Brazil; 4Programa de Pós-graduação em Biologia Parasitária, Instituto Oswaldo Cruz, Rio de Janeiro 21040-360, Brazil; 5Coordenação do Curso de Medicina da Faculdade Metropolitana de Manaus/FAMETRO, Manaus 69050-000, Brazil; 6Faculdade de Farmácia, Universidade Nilton Lins, Manaus 69058-030, Brazil; 7Instituto Leônidas & Maria Deane,-ILMD/FIOCRUZ Amazônia, Manaus 69057-070, Brazil

**Keywords:** *Zika virus*, arbovirus, congenital *Zika virus* syndrome, yellow fever virus, yellow fever virus vaccine, non-microcephalic children

## Abstract

Zika virus (ZIKV) and yellow fever virus (YFV) originated in Africa and expanded to the Americas, where both are co-circulated. It is hypothesized that in areas of high circulation and vaccination coverage against YFV, children of pregnant women have a lower risk of microcephaly. We evaluated the presence and titers of antibodies and outcomes in women who had ZIKV infection during pregnancy. Pregnancy outcomes were classified as severe, moderate, and without any important outcome. An outcome was defined as severe if miscarriage, stillbirth, or microcephaly occurred, and moderate if low birth weight and/or preterm delivery occurred. If none of these events were identified, the pregnancy was defined as having no adverse effects. A sample of 172 pregnant women with an acute ZIKV infection confirmed during pregnancy were collected throughout 2016. About 89% (150 of 169) of them presented immunity against YFV, including 100% (09 of 09) of those who had severe outcomes, 84% (16 of 19) of those who had moderate outcomes, and 89% (125 of 141) of those who had non-outcomes. There was no difference between groups regarding the presence of anti-YFV antibodies (*p* = 0.65) and YFV titers (*p* = 0.6). We were unable to demonstrate a protective association between the presence or titers of YFV antibodies and protection against serious adverse outcomes from exposure to ZIKV in utero.

## 1. Introduction

Zika virus (ZIKV) and yellow fever virus (YFV) are emerging viruses of the flavivirus genus transmitted by mosquitoes of the Aedes genus and are involved in neurological complications and systemic hemorrhagic diseases, respectively [1,2]. ZIKV and YFV originated in Africa and expanded to the Americas, where there is co-circulation of both [1]. ZIKV probably appeared in 1900; however, it was only isolated for the first time in 1947 in *Rhesus macaque* in the Zika Forest, and later in *Aedes africanus* mosquitoes in the same area [3,4]. In humans, it was first isolated in Nigeria in 1953 [5]. The first outbreak in humans occurred in 2007 on the Yap Islands in Micronesia, Pacific Ocean [6]. This virus arrived in the Americas via Brazil in 2013. In February 2016, due to outbreaks and evidence of Guillain–Barré syndrome in adults and congenital Zika virus syndrome (CZVS) in infants born from ZIKV-infected mothers in the Pacific and the Americas, the ZIKV was declared an international public health emergency [7,8]. In addition to CZVS, children born to mothers exposed to the virus may have low birth weight, fetal deaths, spontaneous abortions, alterations in growth and development, and poor growth velocity [9,10,11]. Although it is a disease with important clinical repercussions mainly in pregnant women and their children, there is currently no vaccine for ZIKV [12,13].

YFV originated in Africa and was brought to the Americas through the slave trade, and the first epidemic in the Americas was reported in 1648 in the Yucatán Peninsula. However, the virus was only isolated in a Ghanaian man for the first time in 1927 [14,15]. From viral isolation in the following decade (in 1937), the first vaccine against YFV was produced by the North American Rockefeller Foundation, and from the 1940s, mass campaigns were carried out in South America and vaccination became mandatory in Africa [16,17]. Even with the existence of a vaccine, YFV infection causes significant morbidity and mortality, and is endemic in 44 countries in tropical South America and sub-Saharan Africa, where intermittent large outbreaks among under-vaccinated populations have been recorded [18,19]. Currently, several countries have mass vaccination programs, and in some countries where the disease is endemic, this vaccine is in the national childhood immunization schedule. In countries where the disease is endemic or where there is a yellow fever vaccination program, there is a spectrum of anti-YFV antibodies that may interfere with susceptibility or refractoriness to infections by other flaviviruses due to antigen cross-reactivity and antibody-mediated increase [15,20,21,22].

In type I interferon receptor knockout mice (A129) and BALB/c and SV-129 immunocompetent mice (A129 background), vaccination against YFV provided protection against ZIKV, reduced mortality and cerebral viral load in all mice, and prevented blood loss and loss of weight in BALB/c mice [23]. An in vitro assay showed that ZIKV was capable of infecting BeWo cell lines (derived from human placenta and mimicking the structure and function of the syncytiotrophoblast layer) and HU-VEC (umbilical vein cells), and YFV antibodies presented an impact on ZIKV, depending on the type of cell line. ZIKV was reduced in embryoid bodies in the presence of YFV serum cultured with placental cells, while in direct infection, the embryoid bodies increased viral load in the presence of YFV serum [21]. One study hypothesized that children of pregnant women in regions with high YFV vaccination coverage have a lower risk of microcephaly; however, more appropriate study methodologies should be used [24]. Taken together, these studies suggest that the yellow fever vaccine may be important in the prophylaxis of ZIKV infections and their outcomes [23]. If the YFV vaccine really protects against ZIKV infection and its outcomes, there is a vaccination model that is safe, fast, and cheap, and whose safety is already well known [21,23]. In the present study, we ascertained the presence and titers of antibodies and outcomes in women who had a ZIKV infection during pregnancy and were followed up at a reference hospital for infectious diseases in the Brazilian Amazon, a region with massive immunization against YFV.

## 2. Materials and Methods

### 2.1. Sample Collection and Classification of Outcomes

A sample of pregnant women with an acute ZIKV infection confirmed during pregnancy, who attended the outpatient clinic of the Fundação de Medicina Tropical Doutor Heitor Vieira Dourado (FMT-HVD), Manaus, Amazonas, Brazil, on spontaneous demand between January and December 2016, was evaluated for the presence and levels of anti-YFV antibodies. The demographic and clinical information of the pregnant women and their children was collected and evaluated as previously described [25]. For the purposes of analysis, pregnancy outcomes were classified as severe, moderate, and without outcome, as previously described. Briefly, a severe outcome was defined as miscarriage, stillbirth, or microcephaly occurred. A pregnancy outcome was defined as moderate if low birth weight and/or preterm delivery occurred. If none of these events were identified, the pregnancy was defined as having no adverse effects.

### 2.2. Viral Titer

A viral titer (VT) was necessary for the subsequent Plaque Reduction Neutralization Test (PRNT). VT dosage was performed by counting plaques produced in Vero E6 cell line and expressed in plaque-forming units/mL (PFU/mL). Vero cells were maintained in Dulbecco’s Modified Eagle’s Medium (DMEM, GIBCO^®^, USA) at 28 °C [26]. The growth media were supplemented with 1% fetal bovine serum (FBS, GIBCO^®^, CA, USA), 100 U/mL penicillin, and 100 μg/mL streptomycin (GIBCO^®^, CA, USA). The titration was carried out from the serial dilution of 10–1 to 10–6 of the viral sample, followed by inoculation of 100 μL of the different dilutions in 6-well plates containing Vero cells. The plate was kept in an oven at 37 °C with 5% CO_2_. After discarding the excess virus solution, 2 mL of carboxymethylcellulose medium (CMC, Sigma-Aldrich ^®^, St. Louis, MO, USA) was added. After six days, Vero cells were fixed and stained. The titer was obtained according to the calculation contained in a standardized protocol [26,27].

### 2.3. Plaque Reduction Neutralization Test (PRNT)

Serum samples were inactivated at 56 °C for 60 min and submitted to an assay to determine specific neutralizing antibody titers, according to the assay described in a previously established protocol [26,27]. PRNT was performed in 6-well plates with 80,000 Vero cells/well, using YFV wild-type virus (~50 PFU-) against different serum dilutions (1:20–1:640). The strain was wtYFV#3, isolated in the Laboratório de Flavivirus, FIOCRUZ, Brazil (GenBank: MH018113.1) [28]. The plates were covered with a semi-solid medium (1X DMEM, 1% FBS, 1.5% CMC) and incubated at 37 °C in 5% CO_2_ for six days. After six days, the Vero cells were fixed and stained. PRNT titers were expressed according to 80% plaque inhibition (PRNT80). Titers with PRNT80 < 20 were considered negative [26,27].

### 2.4. Ethical Aspects

The FMT-HVD Research Ethics Committee approved this study, which was assigned the ethical approval number 08941019.2.0000.0005/2019. All participants provided formal written consent.

### 2.5. Statistical Analysis

The variables of interest in this study were registered in a standardized questionnaire in the Epi Info software, version 7. Data analyses were carried out using the “RStudio” program, version 4.2. The results were expressed as relative frequencies, with mean and standard deviation. The groups, based on the outcomes, were compared using the exact fish and chi-square test.

## 3. Results and Discussion

A total of 172 pregnant women sought care at the FMT-HVD between January and December 2016, and their samples were evaluated for the presence and titer of antibodies against the yellow fever virus. The demographics and clinical characteristics of the women are shown in Table 1 according to the outcome and repercussions on the pregnancy and the child. These women, in general, were adults with an average age of 24 years; many were married (44%), had secondary education (55%), and were primiparous (38%). Most women (83.42%) had no outcomes, with the severity being more frequently evidenced in the first trimester of pregnancy (8/9, 89%). Severe outcomes are associated with and described in the literature in women whose infection occurs in the first trimester of pregnancy [25,29].

The high frequency of people with neutralizing antibodies against YFV (150/169, 89%) evidences the presence of immunity against YFV in the region. This is potentially related to the high vaccination coverage or previous infections in the populations of the Amazon region [30]. Anti-YFV antibodies present were present in 100% (09/09) of severe outcomes, 84% (16/19) presents of moderate cases, and in 89% (125/141) of cases with non-outcomes. There was no difference between groups regarding the presence of anti-YFV antibodies (*p* = 0.65) and titers (*p* = 0.6). Most participants in all evaluated groups had titers between 40 and 80 units, and some participants in the unchanged outcome group had titers slightly higher. For the severe outcomes, no woman was seronegative. When comparing seronegative women throughout the stage of pregnancy, we did not find statistically significant differences (*p* = 0.99). These results, therefore, show that there should be no cross-response between YFV and ZIKV or a relationship between the presence of antibodies against YFV and the absence of outcomes. A previous study showed that the presence of antibodies against DENV is not associated with the absence or presence of outcomes in patients with ZIKV infections during pregnancy [29]. This corroborates the definition that there seems to be no cross-immunity between different flaviviruses and ZIKV.

This study, with a transversal design and evaluation of 172 pregnant women from the Amazon region, suggests that there is no association between the presence and high titers of anti-YFV antibodies and protection against serious adverse outcomes in pregnant women exposed to ZIKV, as previously suggested in an ecological study carried out in Brazil on ZIKV [24]. According to these authors, the Northeast region would be more affected by cases of microcephaly due to low vaccination rates for YFV. As we did not find differences in the presence and high titers of antibodies and a reduction in serious outcomes in women exposed to ZIKV, we assume that the differential rates of microcephaly associated with ZIKV between the Northeast and North regions may be associated with different factors, including the onset of the pandemic in the Northeast region; the delay in the registration of cases in the North region, favoring the incorporation of care; and assistance measures in this region that may have helped in the lower frequency of severe events in the North region [31].

This study has limitations. From a descriptive cross-sectional study with a small number of patients, who were not representative of the local or national population, but in whom the presence of high titers of antibodies did not occur as a function of the observed outcomes, we cannot establish a causal relationship since other confounding factors and biases may be involved, such as infections in the first trimester of pregnancy, even with the presence of anti-YFV antibody titers compared to those with mild or no outcome. Furthermore, not all patient samples from previous cohorts were evaluated as they were unavailable [25,31], and the presence of antibodies to other flaviviruses was not tested. Thus, it is suggested that a population-representative cohort study be performed to assess post-vaccination cross-immunity against YFV and other factors in protecting against ZIKV infections and preventing serious outcomes in pregnant women exposed to ZIKV, since few studies exist either to define cross-immunity or to evaluate outcomes.

In conclusion, we were unable to demonstrate a protective association between the presence or the high titers of YFV antibodies and protection against serious adverse outcomes from exposure to ZIKV in utero. However, more robust studies need to be carried out.

## Figures and Tables

**Table 1 viruses-15-02244-t001:** Demographics, clinical characteristics, and viral titers of pregnant women evaluated in the present study according to pregnancy outcomes.

Characteristic	Overall *n* = 169	Severe *n* = 9	Mild *n* = 19	None *n* = 141	*p*-Value
**Civil status**					0.22
Single	67/122 (55%)	5/7 (71%)	6/18 (33%)	56/97 (58%)	
Married	54/122 (44%)	2/7 (29%)	12/18 (67%)	40/97 (41%)	
Divorced	1/122 (0.8%)	0/7 (0%)	0/18 (0%)	1/97 (1.0%)	
Widow	0/122 (0%)	0/7 (0%)	0/18 (0%)	0/97 (0%)	
**Schooling**					0.9
Elementary Incomplete	21/136 (15%)	1/8 (12%)	4/18 (22%)	16/110 (15%)	
Elementary	22/136 (16%)	2/8 (25%)	3/18 (17%)	17/110 (15%)	
Secondary Education	75/136 (55%)	5/8 (62%)	9/18 (50%)	61/110 (55%)	
University Education	18/136 (13%)	0/8 (0%)	2/18 (11%)	16/110 (15%)	
**Parity**					0.51
None	40/104 (38%)	3/4 (75%)	5/14 (36%)	32/86 (37%)	
One	34/104 (33%)	0/4 (0%)	4/14 (29%)	30/86 (35%)	
Two	20/104 (19%)	1/4 (25%)	2/14 (14%)	17/86 (20%)	
More than two	10/104 (6.7%)	0/4 (0%)	3/14 (14%)	7/86 (5.8%)	
**Pregnancy trimester**					<0.001
1°	35/163 (21%)	8/9 (89%)	3/19 (16%)	24/135 (18%)	
2°	59/163 (36%)	0/9 (0%)	8/19 (42%)	51/135 (38%)	
3°	69/163 (42%)	1/9 (11%)	8/19 (42%)	60/135 (44%)	
**Anti-YFV**	150/169 (89%)	9/9 (100%)	16/19 (84%)	125/141 (89%)	0.6
**YFV titer**					
20 to 40	57/169 (34%)	4/9 (44%)	8/19 (42%)	45/141 (32%)	0.6
80 to 160	66/169 (39%)	4/9 (44%)	4/19 (21%)	58/141 (41%)	
320 to 640	18/169 (11%)	1/9 (11%)	2/19 (11%)	15/141 (11%)	
>640	9/169 (5.3%)	0/9 (0%)	2/19 (11%)	7/141 (5.0%)	
**YFV titer per pregnancy trimester**					
*1° Trimester*					>0.99
No titer	1/35 (2.9%)	0/8 (0%)	0/3 (0%)	1/24 (4.2%)	
20 to 40	15/35 (42.9%)	3/8 (38%)	1/3 (33%)	11/24 (46%)	
80 to 160	12/35 (34.3%)	4/8 (50%)	1/3 (33%)	7/24 (29%)	
320 to 640	5/35 (14.3%)	1/8 (12%)	1/3 (33%)	3/24 (12%)	
>640	2/35 (5.7%)	0/8 (0%)	0/3 (0%)	2/24 (8.3%)	
*2° Trimester*					
No titer	10/59 (16.9%)	0/0 (NA%)	2/8 (25%)	8/51 (16%)	0.96
20 to 40	16/59 (27.1%)	0/0 (NA%)	3/8 (38%)	13/51 (25%)	
80 to 160	27/59 (45.8%)	0/0 (NA%)	2/8 (25%)	25/51 (49%)	
320 to 640	4/59 (6.8%)	0/0 (NA%)	0/8 (0%)	4/51 (7.8%)	
>640	2/59 (3.4%)	0/0 (NA%)	1/8 (12%)	1/51 (2.0%)	
*3° Trimester*					
No titer	8/69 (11.6%)	0/1 (0%)	1/8 (12%)	7/60 (12%)	0.43
20 to 40	23/69 (33.3%)	1/1 (100%)	4/8 (50%)	18/60 (30%)	
80 to 160	25/69 (36.2%)	0/1 (0%)	1/8 (12%)	24/60 (40%)	
320 to 640	8/69 (11.6%)	0/1 (0%)	1/8 (12%)	7/60 (12%)	

## Data Availability

Not applicable.

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
