# Peer review of "Does the Presence or a High Titer of Yellow Fever Virus Antibodies Interfere with Pregnancy Outcomes in Women with Zika Virus Infection?"

_viruses, 2023, doi:10.3390/v15112244_

Round 1
Reviewer 1 Report
Comments and Suggestions for Authors
Zika virus (ZIKV) and yellow fever virus (YFV) are belonging to genus flaviviruses. ZIKV emerged on American continent in 2013 and cause public health problems. Particularly, ZIKV was shown to cause congenital ZIKV syndrome. In Brazil, YFV vaccines are available, and thus authors questioned if preexisting anti-YFV immunity in the pregnant women affect outcomes of their children. To understand if cross immunity among flaviviruses affect outcomes of ZIKV infection is quite important.
1. Which strain of YFV did authors use for Neut test? Please clarify.
2. In the Material and Methods, “viral titers” section can be removed.
3. In the reference #25, more than 200 ZIKV cases were confirmed. Why did overall number of ZIKV infections in the prepared manuscript differ? If I understand correctly, authors used same materials.
4. Why do not authors analyze the data of the anti-YFV titers in the each pregnancy trimester?
5. Have authors confirmed that serum used in the study do not possess anti-dengue antibodies?
6. There are several typo. Please check carefully.
Author Response
Dear reviewer, we thank you for your contributions and forward our responses point by point.
- Which strain of YFV did authors use for Neut test? Please clarify.
Source, strain name, gene accession number of the wild-type YFV were added (line 126).
- In the Material and Methods, “viral titers” section can be removed.
Viral titer was necessary for subsequent PRNT. We kept the section and added an explanation in the line 107.
- In the reference #25, more than 200 ZIKV cases were confirmed. Why did overall number of ZIKV infections in the prepared manuscript differ? If I understand correctly, authors used same materials.
The same material was used. But samples from these patients have also been used in other published and ongoing studies. Some ended, leaving only the n sample presented for this work. We added this to the limitations (between lines 190 and 196).
- Why do not authors analyze the data of the anti-YFV titers in each pregnancy trimester?
Values added in table 01.
- Have authors confirmed that serum used in the study do not possess anti-dengue antibodies?
Unfortunately, it has not been confirmed. We add as a limitation (between lines 190 and 196.)
- There are several typo. Please check carefully.
Revised.
Reviewer 2 Report
Comments and Suggestions for Authors
In this study, Piauilino et al investigated whether the presence of antibodies against Yellow Fever virus (YFV) influences the adverse fetal outcomes when the mothers acquire ZIKV infection during pregnancy. The authors showed that the fetal outcomes were not clearly different between 150 YFV-seropositive (89%) and 19 YFV-seronegative (11%) pregnant women, suggesting that the impact of YFV antibodies on the adverse birth outcomes caused by ZIKV is limited. Although it seems difficult to draw a conclusion from this study due to small number of YFV-seronegative samples (19 donors), addressing the association of YFV titers with CZVS would be an important study. I have 1 major comment on this manuscript.
Adverse pregnancy outcomes are strongly associated with the pregnant stages. How many YFV-seronegative women were at 1st, 2nd and 3rd trimester when infected with ZIKV? If the YFV-seronegative women are biased toward later pregnancy compared with YFV-seropositive women, it would become difficult to interpret this study.
Minor comments
1. The sentences lines 72-77 do not inform the reference study (ref. 21) correctly. This study showed that ZIKV infection is reduced in embryoid bodies in the presence of YFV serum when cultured with placental cells (recapitulation of the maternal-fetal interface), while direct infection in embryoid bodies increases viral load in the presence of YFV serum.
2. More detailed information such as source, strain name, gene accession number of the wild-type YFV used for PRNT should be described.
Comments on the Quality of English LanguageSome English grammatical errors that should be corrected were found.
Author Response
Dear reviewer, we thank you for your contributions and forward our responses point by point.
In this study, Piauilino et al investigated whether the presence of antibodies against Yellow Fever virus (YFV) influences the adverse fetal outcomes when the mothers acquire ZIKV infection during pregnancy. The authors showed that the fetal outcomes were not clearly different between 150 YFV-seropositive (89%) and 19 YFV-seronegative (11%) pregnant women, suggesting that the impact of YFV antibodies on the adverse birth outcomes caused by ZIKV is limited. Although it seems difficult to draw a conclusion from this study due to small number of YFV-seronegative samples (19 donors), addressing the association of YFV titers with CZVS would be an important study. I have 1 major comment on this manuscript.
Adverse pregnancy outcomes are strongly associated with the pregnant stages. How many YFV-seronegative women were at 1st, 2nd and 3rd trimester when infected with ZIKV? If the YFV-seronegative women are biased toward later pregnancy compared with YFV-seropositive women, it would become difficult to interpret this study.
The titles were added to table 1 according to outcome and stage of pregnancy. We showed that for the severe outcome, no woman was seronegative. Comparing seronegative women throughout the stage of pregnancy, we did not find statistically significant differences (p=0.99) (Line 168 to 170).
Minor comments
- The sentences lines 72-77 do not inform the reference study (ref. 21) correctly. This study showed that ZIKV infection is reduced in embryoid bodies in the presence of YFV serum when cultured with placental cells (recapitulation of the maternal-fetal interface), while direct infection in embryoid bodies increases viral load in the presence of YFV serum.
Accordingly. We added as suggested.
- More detailed information such as source, strain name, gene accession number of the wild-type YFV used for PRNT should be described.
Accordingly. We added as suggested.
Reviewer 3 Report
Comments and Suggestions for Authors
This brief report presents research examining the relationship between Yellow Fever Virus (YFV) antibodies and pregnancy outcomes in women infected with Zika Virus (ZIKV) during pregnancy in Brazil.
Overall, this is a succinct yet well-written report on an intriguing research question. The background and methods are clearly elucidated. While limited by its design and sample size, the study provides initial evidence that YFV antibodies may not confer protection against adverse ZIKV pregnancy outcomes, contrasting with some prior hypotheses. Further large-scale cohort studies in this population are critical to reach more definitive conclusions.
A brief report is the appropriate level of publication for this study, they had access to a rather limited cohort size but given that the results are contrary to previous studies, it is appropriate to publish with the acknowledgement of those limits.
Author Response
This brief report presents research examining the relationship between Yellow Fever Virus (YFV) antibodies and pregnancy outcomes in women infected with Zika Virus (ZIKV) during pregnancy in Brazil.
Overall, this is a succinct yet well-written report on an intriguing research question. The background and methods are clearly elucidated. While limited by its design and sample size, the study provides initial evidence that YFV antibodies may not confer protection against adverse ZIKV pregnancy outcomes, contrasting with some prior hypotheses. Further large-scale cohort studies in this population are critical to reach more definitive conclusions.
A brief report is the appropriate level of publication for this study, they had access to a rather limited cohort size but given that the results are contrary to previous studies, it is appropriate to publish with the acknowledgement of those limits.
We appreciate the comments. Our study has important limitations, presented in the limitations section (line 190 to 197), but as mentioned, it has important results that are presented in the form of a brief report.
Reviewer 4 Report
Comments and Suggestions for Authors
The manuscript by I. C. Ribeiro Piauilino et al. studied YFV antibody prevalence and compared it to the outcome of ZIKV infection in 169 women.
Mat & Met/ Section 2.2:
This section needs to be revised.
The abbreviation mcVe is strange because it does not correspond to the actual description.
ml or mL; CO2 or CO2 etc.
111: oven = incubator
Line 111: What does it mean that the plates were homogenized every 10 min?
112: onl 2ml of CM or CM in medium?
113: Title is probably titer
108: is it: ...was performed in 10log dilutions ?
122: 1% FBS ?
Table 1: The title of the table should be more meaningful.
42: ZIK = ZIKV
43: Zika Forest or Zika forest in english
21, 38 etc. Zika Virus = Zika virus
33, 38 etc. YFV = Yellow fever virus
Abstract: In the abstract the results should be described more understandable. The % values are confusing and can only be understood after reading Table 1.
The various ZIKV outcomes should be described in more detail and not just referenced.
Sorry, I have been working in the field of virology for over 30 years and have never heard the term antiamaryl. Maybe it is my fault, but a Google search did not yield any result either. Please explain its meaning, as the term appears only as a last comment in the abstract and discussion section. If it is not beneficial to your manuscript, replace it with the term YFV antibody.
Comments on the Quality of English LanguageSome scientific terms need revision.
Author Response
Dear reviewer, we thank you for your contributions and forward our responses point by point.
Mat & Met/ Section 2.2:
This section needs to be revised.
Changed
The abbreviation mcVe is strange because it does not correspond to the actual description.
Changed
ml or mL; CO2 or CO2 etc.
Changed
111: oven = incubator
Changed
Line 111: What does it mean that the plates were homogenized every 10 min?
It is not correct and has been removed
112: onl 2ml of CM or CM in medium?
Changed
113: Title is probably titer
Changed
108: is it: ...was performed in 10log dilutions ?
Changed
122: 1% FBS ?
Changed
Table 1: The title of the table should be more meaningful.
Changed
42: ZIK = ZIKV
Changed
43: Zika Forest or Zika forest in English
Changed
21, 38 etc. Zika Virus = Zika vírus
Changed
33, 38 etc. YFV = Yellow fever virus
Changed
Abstract: In the abstract the results should be described more understandable. The % values are confusing and can only be understood after reading Table 1.
The various ZIKV outcomes should be described in more detail and not just referenced.
Changed
Sorry, I have been working in the field of virology for over 30 years and have never heard the term antiamaryl. Maybe it is my fault, but a Google search did not yield any result either. Please explain its meaning, as the term appears only as a last comment in the abstract and discussion section. If it is not beneficial to your manuscript, replace it with the term YFV antibody.
Thanks for posting. The term was used inappropriately and we made the change as suggested.